# Role of Induced Programmed Cell Death in the Chemopreventive Potential of Apigenin

**DOI:** 10.3390/ijms23073757

**Published:** 2022-03-29

**Authors:** Jung Yoon Jang, Bokyung Sung, Nam Deuk Kim

**Affiliations:** Department of Pharmacy, Research Institute for Drug Development, College of Pharmacy, Pusan National University, Busan 46241, Korea; jungyoon486@hanmail.net (J.Y.J.); auvers1516@gmail.com (B.S.)

**Keywords:** apigenin, apoptosis, autophagy, necroptosis, ferroptosis

## Abstract

The flavonoid apigenin (4′,5,7-trihydroxyflavone), which is one of the most widely distributed phytochemicals in the plant kingdom, is one of the most thoroughly investigated phenolic components. Previous studies have attributed the physiological effects of apigenin to its anti-allergic, antibacterial, antidiabetic, anti-inflammatory, antioxidant, antiviral, and blood-pressure-lowering properties, and its documented anticancer properties have been attributed to the induction of apoptosis and autophagy, the inhibition of inflammation, angiogenesis, and cell proliferation, and the regulation of cellular responses to oxidative stress and DNA damage. The most well-known mechanism for the compound’s anticancer effects in human cancer cell lines is apoptosis, followed by autophagy, and studies have also reported that apigenin induces novel cell death mechanisms, such as necroptosis and ferroptosis. Therefore, the aim of this paper is to review the therapeutic potential of apigenin as a chemopreventive agent, as well as the roles of programmed cell death mechanisms in the compound’s chemopreventive properties.

## 1. Introduction

Cancer is a globally important health issue and is the second leading cause of death in the United States, where 1,898,160 new cancer cases and 608,570 cancer deaths were expected in 2021. Cancer mortality increased during the 20th century but decreased by 31% from 1991 to 2018 [1]. However, even though cancer treatment has been advanced significantly over the past two decades (e.g., development of safer, more effective, and more precise drugs) and molecular approaches have been used to treat neoplasms and reduce mortality, the side effects of such treatments remain a major problem [2], and some cancer cells develop resistance or evasion mechanisms [3]. Furthermore, modern cancer treatments are expensive. Therefore, chemoprevention is attracting increasing attention as a cheaper and more effective strategy for reducing cancer-related mortality [2].

Chemopreventive strategies involve the use of natural, synthetic, or biological agents to prevent, inhibit, or reverse the early stages of carcinogenesis or to prevent invasion by premalignant cells. Natural compounds may also reduce side effects [4]. Clinically, these strategies are classified as primary, secondary, or tertiary and are used to reduce the risk of cancer incidence in high-risk populations, to reduce the progression of cancer (via drug treatment) in patients with premalignant lesions, and to prevent cancer recurrence, respectively [5,6]. The definitions of primary and secondary chemoprevention change, and some researchers do not distinguish between primary and secondary chemoprevention. However, typical examples of primary chemoprevention agents include dietary phytochemicals and nonsteroidal anti-inflammatory drugs.

Recently, both herbal and phytochemical-based medicines have attracted attention for their effectiveness against cancer, as well as a wide variety of other diseases [7,8,9,10,11,12,13,14,15]. Indeed, researchers around the world are focusing on the chemopreventive, antioxidant, and anti-inflammatory properties of bioactive compounds [16,17,18], and natural products and their derivatives account for one-third of all new drugs approved by the United States Food and Drug Administration (FDA) [19,20,21]. Plant-based medicines contain multiple bioactive compounds (e.g., alkaloids, carotenoids, diterpenoids, flavonoids, phenolic compounds, and tannins) that impart unique medicinal properties [22,23], and accordingly, plant-derived compounds play important roles in increasing the sensitivity of cells to standard chemotherapy and in reducing cancer risk, invasion, and metastasis [22,24,25,26].

The naturally occurring flavonoid apigenin (4′,5,7-trihydroxyflavone), in particular, which is one of the most widely distributed phytochemicals in the plant kingdom, is one of the most thoroughly researched phenolic compounds [27]. The compound has very low toxicity, is abundant in fruits and vegetables, has many potential biological activities, including anticancer effects, and can simultaneously exert multiple anticancer effects through the modulation of important molecular targets [28,29]. The aim of this paper is to review the therapeutic potential of apigenin as a chemopreventive agent, as well the roles of programmed cell death (PCD) mechanisms in the compound’s chemopreventive properties.

## 2. Apigenin

The common name of apigenin (i.e., 4′,5,7-trihydroxyflavone; C_15_H_10_O_5_, 270.24 g/mol) is derived from the genus *Apium* (Apiaceae or Umbelliferae). The yellow crystalline compound possesses hydroxyl groups at the C-5 and C-7 positions of the A-ring and at the C-4′ position of the B-ring and is insoluble in water but soluble in dimethyl sulfoxide and hot ethanol [5,30].

Apigenin is considered an important flavonoid, due to its abundance in a variety of natural sources, including fruits and vegetables, and major sources include parsley, chamomile, celery, spinach, artichoke, and oregano. Dried parsley contains 45,035 μg/g of apigenin, whereas chamomile (dried flowers), celery seed, vine spinach, and Chinese celery contain 3000–5000, 786.5, 622, and 240.2 µg/g, respectively. Glycosylated derivatives (e.g., apiin and apigetrin) and dimers (e.g., amentoflavones, such as 3′,8″-biapigenin) of apigenin have also been isolated from natural sources [5].

## 3. Physiological Functions of Apigenin

Apigenin has been used in traditional medicines, owing to its anti-inflammatory and antioxidant [29,31], blood-pressure-lowering [32], antibacterial and antiviral [33], antidiabetic [34], and anti-allergic properties [35]. Recently, apigenin has also been demonstrated to possess tumor-suppressive effects, and since Birt et al. [36] first reported the anticancer activity of apigenin in 1986, the compound has been reported to exert anti-tumor effects in a variety of cancer types in both in vitro cell lines and in vivo mouse models (Figure 1).

## 4. Apigenin in Cancer Therapy

Carcinogenesis is a multi-step process that involves a series of genetic and epigenetic changes that contribute to the initiation, promotion, and development of cancer [37,38,39]. Cancer treatment strategies include the induction of cell apoptosis to eradicate tumor cells and the induction of cell cycle arrest to prevent cancer cell proliferation, thereby prolonging patient survival [40,41,42], and strategies involving the promotion of apoptosis/autophagy, control of the cell cycle, prevention of tumor cell migration and invasion, and induction of patient immune responses have also been proposed [43,44,45,46].

Apigenin has been demonstrated, in both in vitro and in vivo models, to exert broad anticancer effects in a variety of cancer types, including colorectal cancer, breast cancer, liver cancer, lung cancer, melanoma, prostate cancer, and osteosarcoma [47,48,49,50,51,52]. The compound can prevent cancer cell proliferation by triggering apoptosis, which leads to autophagy and cell cycle regulation, and can also reduce cancer cell motility, thereby preventing cancer cell migration and invasion. It was also recently reported that apigenin can inhibit cancer by stimulating patient immune response [53] and that the compound can regulate several protein kinases and signaling pathways, including PI3K/AKT, MAPK/ERK, JAK/STAT, NF-κB, and Wnt/β-catenin pathways [28].

## 5. Effect of Apigenin on Apoptosis

### 5.1. Apoptosis

The term apoptosis was first used by Kerr et al. [54] in 1972 to describe a morphologically distinct type of cell death. Apoptosis, or Type I PCD, is a closely linked cellular process that plays an important role in the development and homeostasis of multicellular organisms [5]. Because tissue homeostasis involves a balance between apoptosis and cell proliferation, disruption of this balance (e.g., uncontrolled apoptosis) may be implicated in a variety of human diseases, including cancer [55,56]. Apoptosis is mainly induced through the intrinsic (mitochondrial) and extrinsic (death receptor) pathway.

### 5.2. Types of Apoptosis

#### 5.2.1. Intrinsic (Mitochondrial) Pathway

The intrinsic pathway, also known as the mitochondrial pathway of apoptosis, involves various stimuli that act on multiple cellular targets within the cell. This form of apoptosis depends on factors that are released from the mitochondria and begins in either a positive or a negative pathway. Negative signals are caused by the absence of cytokines, hormones, and growth factors in the cell’s immediate environment. In the absence of these survival signals, pro-apoptotic molecules within cells, such as Bax, Noxa, and the p53-upregulated modulator of apoptosis (PUMA), which are normally restrained, are activated to initiate apoptosis. Other factors initiating apoptosis are positive and include exposure to viruses and various toxic substances, radiation, hypoxia, reactive oxygen species (ROS), and toxins [57].

The intrinsic apoptotic pathway is controlled by the mitochondria, including key apoptotic factors, such as cytochrome *c* [58]. The intrinsic pathway is also controlled by the members of the Bcl-2 family. Pro- and anti-apoptotic Bcl-2 proteins are localized in mitochondria to manage the release of apoptogenic factors [59]. The pro-apoptotic Bcl-2 protein induces permeability of the outer mitochondrial membrane, allowing cytochrome *c* to be released from the mitochondrial intermembrane space [60]. Consequently, in the presence of ATP, it binds to apoptotic protease activating factor 1 (Apaf-1) and participates in the formation of a multimeric Apaf-1/cytochrome *c* complex. Subsequently, the Apaf-1/cytochrome *c* complex binds to procaspase-9 to generate an apoptosome [61]. Consequently, procaspase-9 is cleaved, activated, and dissociated from the apoptosome. Once activated, caspase-9 is activated by cleaving executive caspase-3, -6, and/or -7 [62].

#### 5.2.2. Extrinsic (Death Receptor) Pathway

The extrinsic apoptotic pathway relies on cell surface death receptors, such as tumor necrosis factor (TNF), which are controlled by the expression levels of triggering ligands [63,64,65]. Ligands that stimulate cell surface death receptors contain cytokines, such as transforming growth factor beta 1 (TGF-β1), TNF-α, and interferon gamma [65]. Cell surface death receptors initiate procaspases via ligand binding [66]. Death domains play an important role in the transduction of death signals from the cell surface to intracellular signaling pathways. Therefore, when cell surface death receptor–ligand binding occurs, cytoplasmic adapter proteins are recruited and associated with procaspase-8 via dimerization of the death effector domain [65]. Next, a death-inducing signaling complex (DISC) is formed, which triggers the autocatalytic activation of procaspase-8. Once activated, caspase-8 prompts executioner caspases, such as caspase-3, -6, and -7, which mediate the execution stage of apoptosis [65,67].

### 5.3. Induction of Apoptosis by Apigenin

The modulation of apoptosis has significant implications for cancer therapy, and thus, the effects of apigenin on molecular targets have attracted extensive investigation (Table 1).

#### 5.3.1. Effect of Apigenin on Caspase-Mediated Apoptosis

Caspases are a family of cysteine proteases that provide important connections in the cellular networks that control inflammation and apoptosis. More than 12 caspases have been reported to date, and caspase-2, -3, -6, -7, -8, -9, and -10 have been implicated in apoptosis. Depending on their mechanism of action, these enzymes are broadly categorized as initiator caspases (caspase-2, -8, -9, and -10) and effector (or executioner) caspases (caspase-3, -6, and -7). Initiator caspases activate effector caspases, which modulate their activity to destroy key structural proteins and to activate other enzymes. In addition, caspase activation is mediated by both intrinsic and extrinsic pathways. Therefore, caspase function and expression are downregulated in tumors, which suggests that caspase activation may be an effective strategy for cancer treatment [151].

The ability of apigenin to induce caspase activation and caspase-dependent apoptosis has been demonstrated in cell lines associated with a variety of cancer types, including bladder cancer, breast cancer, cervical cancer, colon cancer, esophageal cancer, gastric cancer, glioblastoma, head and neck cancer, melanoma, leukemia, liver cancer, lung cancer, mesothelioma, neuroblastoma, osteosarcoma, ovarian cancer, pancreatic cancer, prostate cancer, renal cancer, and thyroid cancer. For example, many studies have demonstrated the apoptotic effect of apigenin, via caspase activation, on breast cancer cells. In MDA-MB-453 breast cancer cells, apigenin activates caspase-8, -9, and -3 and causes the cleavage of poly(ADP-ribose) polymerase (PARP), which results in apoptosis [82,83], and apoptosis is also induced by apigenin-mediated caspase-3 activation in MDA-MB-231, BT-474, SKBR3, T47D, and HBL-100 breast cancer cells [71,72,73,74,75,76,77,80]. Seo et al. [80] reported that extrinsic caspase-dependent apoptosis upregulates levels of cleaved caspase-8 and -3 in apigenin-treated BT-474 breast cancer cells, and consequently, the induction of PARP cleavage was confirmed. In addition, treatment with the caspase-8 inhibitor Z-IETD-FMK and the caspase-9 inhibitor Z-LEHD-FMK, together with apigenin, induced caspase-dependent apoptosis in BT-474 cells, and apigenin has been reported to trigger apoptotic cell death in caspase-3-deficient MCF-7 cells [152]. This can be demonstrated by the activation of caspase-8 by apigenin, which results in proteolytic cleavage of PARP [72,77,78,79]. Furthermore, using the caspase-9-specific inhibitor Z-LEHD-FMK and the general caspase inhibitor Z-VAD-FMK, apigenin was confirmed to induce apoptosis caspase-dependent apoptosis in PC3 and DU145 cells in a dose-dependent manner [153]. A similar effect was demonstrated in 22Rv1 human prostate cancer epithelial cells treated with apigenin, using the general caspase inhibitor Z-VAD-FMK [140]. Das et al. [100] demonstrated that reduced cytochrome *c* levels, owing to apigenin-induced increases in caspase-3 and -9 levels, induce apoptosis in A375 melanoma cells. Furthermore, apigenin with poly(lactide-co-glycolide)-containing nanoparticles was reported to improve the regulation of cell death and cytochrome *c* release and the expression of Apaf-1, Bax, Bcl-2, caspase-9, caspase-3, and PARP cleavage in A375 cells [101], and apigenin nanoparticles have been reported to contribute to the inhibition of ultraviolet (UV)-B-induced skin tumor growth by inducing caspase-3-mediated apoptosis [154].

#### 5.3.2. Effect of Apigenin on Tumor Suppressor p53-Dependent Apoptosis

The tumor suppressor protein p53 is a transcription product of the anti-oncogene *TP53* and is an important factor in the termination of cellular cancerization and induction of apoptosis in cancer cells. As such, p53 is described as the “guardian of the genome” [155]. The ability of p53 to regulate apoptosis is one of the most widely studied areas, and studies have shown that apoptosis contributes to the tumor-suppressive activity of p53. As a proapoptotic mediator, p53 can activate the transcription of proapoptotic genes, and p53 includes BH-3-specific proteins that encode members of the Bcl-2 family, such as Bax, Noxa, and PUMA. However, p53 may also promote caspase activation by inhibiting anti-apoptotic genes, such as *survivin*, upregulating apoptosis-inducing gene products, including Fas, TRAIL receptor DR5, Bid, and Apaf-1 [156]. Torkin et al. [130] reported that apigenin induces apoptosis in human neuroblastoma cells but not in untransformed cells. The action of apigenin appears to be mediated by p53, since it increases the levels of p53 and p53 target genes, *p21^WAF1/CIP1^* and *Bax*. Furthermore, apigenin-mediated apoptotic cell death has been reported to occur in wild-type p53 cells, but not in non-functional mutant p53 cells. Shukla et al. [140] used p53 antisense oligonucleotide experiments to demonstrate that a p53-associated pathway is required for apigenin-mediated apoptosis. In prostate cancer 22Rv1 cells, apigenin treatment increased the expression and transcriptional activation of p53. Therefore, increased p53 protein expression correlated with an increase in the level of the transcriptional target *p21^WAF1/CIP1^*. Moreover, consistent with in vitro findings, the uptake of apigenin by 22Rv1-transplanted nude mice was reported to increase wild-type p53, p53-Ser15 phosphorylation, cytochrome *c*, and cleaved caspase-3 expression in a dose-dependent manner, and the resulting up- and down-regulation of Bax and Bcl-2 levels, respectively, suggest that the inhibited growth of 22Rv1 tumor xenografts is due to the induction of p53 pathway-mediated apoptosis. According to Shendge et al. [152], the apoptosis of apigenin-treated MCF-7 cells involved increased p53 expression, Bax/Bcl-2 ratio, caspase activation, and PARP cleavage. Meanwhile, treatment with both apigenin and the p53-mediated apoptosis inhibitor pifithrin-μ reduced the apoptotic cell population, thereby revealing the important role of p53 in apigenin-induced apoptosis in MCF-7 cells.

#### 5.3.3. Effect of Apigenin on Tumor Suppressor p53-Independent Apoptosis

Mutations in p53 have been identified in more than 50% of human tumor tissues. In certain tumor types, the loss of p53 function is associated with chemoresistance, and cancers with p53 mutations generally respond poorly to therapeutics [157], thereby prompting the investigation of anticancer agents that act independently of p53 status. Zhang et al. [94] reported that apigenin induced apoptosis in p53 mutants of human esophageal squamous cell carcinoma KYSE-510 cells via the mitochondrial apoptosis pathway and induction of p21^WAF1/CIP1^. Meanwhile, in prostate cancer cells, DU145 (with mutated p53) and PC-3 (with null p53), apigenin treatment increased p21^WAF1/CIP1^ expression and induced apoptosis. These results demonstrate that apigenin exerts a p53-independent chemopreventive effect [141,142,143,144,145,146,147,148]. King et al. [135] reported that, in human pancreatic cancer cells (BxPC-3 and MIA PaCa-2), the p53 DNA binding-specific inhibitor pipitrin-α blocked transcription-dependent p53 activation and, thus, apigenin’s anti-proliferative and pro-apoptotic effects. Even though there was little reversal of this effect, the p53-regulated apoptosis p21^WAF1/CIP1^ and PUMA was inhibited by pifithrin-α. Therefore, apigenin can activate p53 through a parallel and transcriptionally independent pathway of PCD.

## 6. Effect of Apigenin on Autophagy

### 6.1. Autophagy

Autophagy, or Type 2 PCD [158], is characterized by the sequestration of cytoplasmic material into vacuoles for mass degradation by lysosomal enzymes and is defined as the cellular process through which cytoplasmic macromolecules and organelles are delivered to lysosome for degradation [159]. Much evidence supports the hypothesis that autophagy has a complex and contradictory relationship with cancer. [160] During starvation, autophagy provides recycled metabolic substrates and may promote cell survival by maintaining energy homeostasis. However, autophagy can either cooperate with apoptosis or trigger apoptosis as a backup mechanism [161]. Autophagy involves a variety of proteins that are encoded by autophagy-related genes (ATGs), of which more than 30 have been reported. In general, autophagy is induced by the activation of AMP-activated protein kinase (AMPK), which results from a lack of energy in the form of ATP. However, the process is also negatively regulated by mammalian target of rapamycin (mTOR), and the activation of mTOR complex 1 (mTORC1) has been reported to prevent autophagy, whereas its inhabitation has been reported to trigger autophagy when growth factors and/or amino acids are insufficient [162].

### 6.2. Types of Autophagy

The main types of autophagy (i.e., microautophagy, macroautophagy, and chaperone-mediated autophagy (CMA)) are characterized by their functions and by the way cargo is delivered to lysosomes [163]. The most well-known type, macroautophagy, involves the formation of double-membrane vesicles (i.e., autophagosomes) that swallow other vesicles (such as proteins, mitochondria, and peroxisomes) and fuse with other lysosomes and lysosomal hydrolases to degrade their contents [164]. Meanwhile, microautophagy is a non-selective lysosomal degradation process by which cytoplasmic cargo are engulfed directly from the boundary membrane via autophagy tubes that mediate endoluminal incorporation and vesicle cleavage [165], and CMA, which only occurs in mammalian cells, differs from other forms of autophagy in both the way transport proteins are perceived for lysosome transfer and the way these proteins reach the lysosomal lumen. In CMA, the internalization of substrate proteins precedes deployment, a step that is not necessary for other types of autophagy [166]. Several recent studies have highlighted the significant role of microautophagy and CMA in tumor growth and progression. However, nearly all studies of the role of autophagy in cancer development, progression, and treatment refer to macroautophagy [5].

### 6.3. Induction of Autophagy by Apigenin

The diverse molecular targets of apigenin-induced autophagy are summarized in Table 2. The induction of non-apoptotic autophagy by apigenin treatment was first reported by Ruela-de-Sousa et al. [106] in erythroleukemia TF1 cells, in which the autophagy inhibitor mTOR and its downstream 70-kDa ribosomal protein S6 kinase (p70S6K) were inhibited. The treatment failed to affect beclin 1 levels but strongly reduced Atg5, 7, and 12 and induced the production of both non-electron-dense vacuoles and double-membrane vacuoles, which constitute strong evidence of TF-1 cell autophagy. Subsequent studies have confirmed that apigenin can induce autophagy and have reported that apigenin can function as either a tumor suppressor or protector [5,167]. In one study [89], apigenin-induced autophagy was characterized by an increase in the level of LC3-II, which is a processing form of LC3-I, the appearance of autophagosomes, and the accumulation of acid vesicles. In addition, the autophagy inhibitor 3-methyladenine (3-MA) significantly enhanced apigenin-induced apoptosis, with increased levels of PARP cleavage, but reversed apigenin-induced LC3 puncta, which suggested that apigenin induced apoptosis and autophagy simultaneously and that apigenin-induced autophagy plays a cytoprotective role in apigenin-caused apoptosis. Similarly, Yang et al. [111] reported that apigenin increased the expression of LC3-II and the number of GFP-LC3 puncta in HepG2 cells. In addition, it has been reported that the inhibition of autophagy by 3-MA and Atg5 gene silencing enhances the apigenin-induced inhibition of proliferation and apoptosis and that apigenin induces both apoptosis and autophagy by suppressing the PI3K/Akt/mTOR pathway. Most importantly, in vivo data demonstrate that apigenin can reduce tumor growth, and the inhibition of autophagy by 3-MA notably enhances the anticancer effect of apigenin. Chen et al. [92] reported that apigenin induces autophagy and apoptosis in cisplatin-resistant colon cancer cells by inhibiting the m-TOR/PI3K/AKT signaling pathway, increases levels of the autophagy-related proteins Beclin-1 and LC3-II, and inhibits p62 expression. In vivo data have also demonstrated that apigenin can inhibit tumor growth in xenografted mouse models.

According to Kim et al. [168], apigenin treatment increases the phosphorylation of ATG5, LC3-II, AMPK, and ULK1 and downregulates p62, thereby promoting autophagic cell death, in gastric cancer AGS and SNU-638 cell lines under hypoxic conditions. Apparently, apigenin can also induce autophagic cell death by activating protein kinase R-like endoplasmic reticulum kinase (PERK) signaling, which is indicative of the endoplasmic reticulum (ER) stress response, and induces ER stress and autophagy-related apoptosis by inhibiting hypoxia-inducible factor 1, alpha subunit (HIF-1α), and enhancer of zeste homolog 2 (Ezh2) under both normoxic and hypoxic conditions. Therefore, apigenin clearly activates autophagic cell death by suppressing HIF-1α and Ezh2 in gastric cancer cells under hypoxic conditions.

**Table 2 ijms-23-03757-t002:** Molecular targets of apigenin-induced autophagy.

Cancer/Cell Lines	Up-Regulation	Down-Regulation	Refs.
** *Breast* **			
T47D and MDA-MB-231	LC3-I, LC3-II		[75]
** *Cevical* **			
HeLa		GRP78	[169]
** *Colon* **			
HCT116	LC3-II	Wnt, c-Myc, Axin2, cyclin D1, β-catenin, p-AKT, p70S6, p-p70, S6, 4EBP1, p-4EBP1	[89,170]
SW480	LC3-II	Wnt	[170]
HT-29	Beclin-1, LC3-II	p62, p-mTOR, p-PI3K, p-AKT	[92]
** *Gastric* **			
AGS and SNU-638	Atg5, Beclin1, LC3-II AMPKα ULK1, GRP78, p-PERK, p-eIF2α ATF4, CHOP, GRP78, CD63	p62, p-mTOR, Ezh2	[168]
** *Liver* **			
HepG2 and HepG2 xenograft	LC3-I, LC3-II, Atg5, Beclin1, LC3-II/I ratio, AMPK	SQSTM1/p62, p-PI3K, p-AKT, p-mTOR, p-mTOR/mTOR ratio, NQO2	[111,171,172,173]
Hep3B	LC3-II, Atg7, ROS		[115]
SMMC-7721 and SK-HEP-1	LC3B-II, ULK1	p62	[174]
** *Leukemia* **			
TF-1	LC3-II, Atg5, Atg12, LMWPTP	p-Src, p-JAK2,p-STAT3, p-STAT5, p-SHP2, p-mTOR, p-p70S6K	[106]
** *Lung* **			
H1975	LC3-II	p-EGFR, Kras, c-Myc, HIF-1α, p-AMPKα	[175]
** *Multiple myeloma* **			
NCI-H929	Beclin1, LC3B-II		[176]
** *Neuroblastoma* **			
SH-SY5Y	LC3-II, p-AKT, mTOR	Beclin 1, TLR-4, Myd88	[177]
** *Pancreatic* **			
PANC-1	LC3-I, LC3-II, p-AKT	p62, NRF2, SOD, CATALASE, HSP90, p-4EBP1	[178]
PaCa-44	LC3-I, LC3-II, p62, NRF2, SOD, catalase, HSP90, 4EBP1, p-AKT		[178]
** *Renal* **			
ACHN and OS-RC-2	Beclin1, LC3-II, p-AMPKα, p-JNK	Ki-67, PCNA, p62, p-PI3K, p-AKT, p-mTOR	[179]
** *Skin* **			
COLO-16 and HEK	ATM, ATR, UPR, BiP, IRE1α, PERK, Atg, LC3-I, LC3-II		[180]
** *Thyroid* **			
BCPAP	Beclin1, LC3-I, LC3-II, Nrf2, HO-1	p62	[181]

AMPK, 5′ adenosine monophosphate-activated protein kinase; ATR, ATF4, activating transcription factor 4; ATR, ataxia telangiectasia and Rad3-related protein; ATM, ataxia-telangiectasia mutated; Atg5, autophagy-related 5; Atg7, autophagy-related 7; Atg12, autophagy-related 12; Axin2, axis inhibition protein 2; CHOP, C/EBP homologous protein; 4EBP1, eukaryotic translation initiation factor 4E binding protein 1; EGFR, epidermal growth factor receptor; Ezh2, enhancer of zeste homolog 2; GRP78, binding immunoglobulin protein; HIF-1α, hypoxia-inducible factor 1-alpha, HO-1, heme oxygenase-1; Hsp90, heat shock protein 90; IRE1α, inositol requiring transmembrane kinase endoribonuclease-1α; JAK2, Janus kinase 2; LMWPTP, low-molecular-weight protein tyrosine phosphatase; mTOR, mammalian target of rapamycin; MYD88, myeloid differentiation primary response 88; Nrf2, nuclear factor erythroid 2-related factor 2; NQO2, NRH-quinone oxidoreductase 2; p70S6K, 70-kDa ribosomal protein S6 kinase; PI3K, phosphoinositide 3-kinase; PCNA, proliferating cell nuclear antigen; PERK, protein kinase RNA-like endoplasmic reticulum kinase; ROS, reactive oxygen species; SHP2, Src homology region 2 domain-containing phosphatase-2; SOD, superoxide dismutase; STAT, signal transducer and activator of transcription; TLR-4, Toll-like receptor 4; ULK1, autophagy-activating kinase 1.

## 7. Effect of Apigenin on Necroptosis

### 7.1. Necroptosis

Necroptosis is a novel form of PCD with morphological features similar to necroptosis, as described by Degterev et al. [182] in 2005. Necroptosis has several features, such as apoptosis and necrosis. For example, morphological signs, such as increased cell size, expanded organelles, translucent cytoplasm, premature plasma membrane destruction, and apoptosis, can be reversed [183]. Even though necroptosis plays an important role in the efficacy of several cancer therapeutics, several signaling pathways have been implicated in the activation of necrosis [184,185]. Necroptosis is a caspase-independent process that is involved in the activation of death receptors [186]. During necroptosis, substrate mixed-lineage kinase domain-like (MLKL)/receptor-interacting serine/threonine kinase 3 (PIRK3) plays an important role in the activation and execution of cell death [187]. After the phosphorylation of MLKL by PIRK3, MLKL is oligomerized and translated into the plasma membrane, where it improves membrane permeability by interacting with phospholipids. Permeability is the main difference between apoptosis and necrosis. To further characterize necroptosis, as well as the difference between apoptosis and necrosis, apoptotic cells are surrounded by adjacent cells or antigen-presenting cells, whereas in necroptosis, permeability increases the release of cytokines and chemokines to induce immune responses and inflammation [188].

### 7.2. Necroptosis in Cancer

Necroptosis has been described as both a friend and an enemy of cancer and has been reported to exert this dual effect on the growth of tumors associated with various types of cancers. As an unsafe form of cell death that occurs in non-apoptotic cells, necroptosis can stop tumor development. Nevertheless, as a form of necrotic cell death, necroptosis can induce inflammatory responses and has been reported to promote cancer metastasis and immunosuppression [189,190]. Therefore, the manipulating and/or induction of necroptosis in anticancer therapy represent promising therapeutic approaches that could bypass acquired or intrinsic apoptosis resistance and serve as alternatives for eliminating apoptosis-resistant cancer cells. A growing number of compounds and chemotherapeutic agents have been reported to induce necroptosis in cancer cells [191].

### 7.3. Induction of Necroptosis by Apigenin

Even though few studies have investigated the role of apigenin in necroptosis, the several molecular targets of apigenin-induced necroptosis are summarized in Table 3. Necroptosis involves activation of receptor-interacting protein kinase (RIPK) 1, which binds to RIPK3 to form a necrosome. These events ultimately activate mixed-lineage kinase domain-like protein (MLKL), which causes necroptosis [192]. Lee et al. [193] reported that apigenin treatment can increase p-MLKL and p-RIP3 levels in malignant mesothelioma cell lines (MSTO-211H and H2452) and that apigenin can significantly inhibit cell viability, increase ROS, and induce ATP depletion through mitochondrial dysfunction, thus promoting ROS-dependent necroptosis. Meanwhile, Warkad et al. [194] reported that combined treatment with metformin and apigenin upregulates the necroptosis-related factors p-MLKL and p-RIP3 in AsPC-1 pancreatic cancer cells and that metformin and apigenin together, but not individually, can dramatically increase ROS levels and reduce cell viability in a variety of cancer cells, including AsPC-1 cells. Warkad et al. also reported that metformin differentially regulates cellular ROS levels through the AMPK-FOXO3a, forkhead box O3a (FOXO3a)-MnSOD pathways in AsPC-1 pancreatic cancer cells and that the combination of metformin and apigenin induces DNA damage by AsPC-1 pancreatic cancer cell-specific ROS amplification, which results in apoptosis, autophagy, and necroptosis.

## 8. Effect of Apigenin on Ferroptosis

### 8.1. Ferroptosis

Ferroptosis, which was first reported by Dixon et al. [195] in 2012, is a form of apoptosis characterized by intracellular iron accumulation and the cellular accumulation of lipid ROS. The process can be stimulated by ROS generation, GSH depletion, and nicotinamide adenine dinucleotide phosphate (NADPH)-dependent lipid peroxidation [196] and involves mitogen-activated protein kinases (MAPKs), including c-Jun NH2-terminal kinase (JNK), ERK, and p38 [162]. The morphological features of ferroptosis include increased mitochondrial membrane density, reduced mitochondrial crista and mitochondrial size, and mitochondrial exoplanet rupture, possibly owing to the dysfunction of voltage-dependent anionic channels and changes in mitochondrial membranes fluidity via lipid peroxidation [197,198].

### 8.2. Ferroptosis and Cancer

Cell death is important for homeostasis, normal development, and the prevention of hyperproliferative diseases, such as cancer. Despite the success of clinical cancer treatment, genetic resistance to conventional chemotherapeutic agents remains problematic [199]. Ferroptosis has been used to treat a variety of physiological and pathological processes and diseases, including several types of cancer. Many studies have reported that ferroptosis plays an important role in killing tumor cells and preventing tumor growth. For example, ferroptosis has been reported to inhibit tumorigenic cells associated with hepatocellular carcinoma [200], leukemia [201], non-small-cell lung cancer [202], pancreatic cancer [203], and breast cancer [204]. Therefore, ferroptosis could be used as a novel therapeutic strategy for cancer treatment. Because several FDA-approved clinical drugs (e.g., artesunate, sorafenib, and sulfasalazine) are known to induce ferroptosis in certain types of cancer, ferroptosis can be used in preclinical and clinical studies. Moreover, ferroptosis-inducing agents, such as erastin, piperazine erastin, and RSL3, have been reported to inhibit tumor growth in xenograft models of HT-1080 cells in vivo [196]. Therefore, there is a need for clinical studies of ferroptosis-inducing drugs for use in tumor therapy [205].

### 8.3. Induction of Ferroptosis by Apigenin

Few studies have investigated the effects of apigenin on ferroptosis. Therefore, a limited numbers of molecular targets of apigenin-induced ferroptosis are summarized in Table 4. According to Adham et al. [176], apigenin treatment can induce cell cycle arrest, apoptosis, autophagy, and ferroptosis in the multiple myeloma cell line NCI-H929. Apigenin-induced ferroptosis was confirmed by treating NCI-H929 cells with apigenin and the ferroptosis inhibitor ferrostatin-1, which completely ameliorated apigenin’s cytotoxicity. Meanwhile, another ferroptosis inhibitor, namely, deferoxamine, reduced the cytotoxicity of apigenin by 3.1-fold. In addition to providing the first evidence that apigenin is involved in ferroptosis, Adham et al. also demonstrated that apigenin is an important contributor to the inhibition of the STAT1/COX-2/iNOS signaling pathway to inhibit inflammation and induce apoptosis and that apigenin may be a suitable candidate for treating multiple myeloma. Shao et al. [206] reported that myeloperoxidase (MPO)-mediated oxidative stress plays an important role in pathological dysfunction and also demonstrated that apigenin can relieve MPO-mediated oxidative stress and inhibit neuronal ferroptosis, thereby significantly increasing GPX4, an important marker of ferroptosis. Liu et al. [207], who investigated mesoporous magnetic nanosystems for apigenin (API) delivery, reported that the targeted Fe_2_O_3_/Fe_3_O_4_@mSiO_2_-HA nanocomposite delivery system significantly increased ROS levels and cellular lipid peroxidation levels, which is typical of ferroptosis in A549 cells, and upregulated COX2 and p53, an important gene in ferroptosis, while also downregulating GPX4 and FTH1. The downregulation of GPX4, which is also an important component of the ferroptosis signaling pathway, which involves iron ions. The simultaneous administration of apigenin and the ferroptosis inhibitor ferrostatin-1 was reported to yield less pronounced cell inhibitory effects than the administration of apigenin alone. Adham et al. [176] reported that extracts of *Thymus vulgaris* and *Arctium lappa* induced apoptosis, autophagy, and ferroptosis in leukemia and multiple myeloma cell lines, and apigenin has been identified in *T. vulgaris*. In a multiple-myeloma cell line (NCI-H929), *T. vulgaris* and *A. lappa* extracts neutralized cytotoxic activity up to the highest concentration of the experiment (100 μg/mL) using the ferroptosis inhibitors ferrostatin-1 and deferoxamine.

## 9. Conclusions

This paper reviews the chemopreventive effects of apigenin and the roles of apoptosis, autophagy, necroptosis, and ferroptosis in the compound’s physiological effects. Evidence from both in vitro and in vivo studies indicates that apigenin exerts significant anticancer activity. However, even though apigenin is bioavailable after oral administration in rats and mice, there are no data regarding the compound’s pharmacodynamic or pharmacokinetic profiles in humans. Therefore, additional data, including the bioavailability and safety of apigenin in humans, are needed to promote further investigation and the development of apigenin as a chemopreventive or therapeutic anticancer agent.

## Figures and Tables

**Figure 1 ijms-23-03757-f001:**
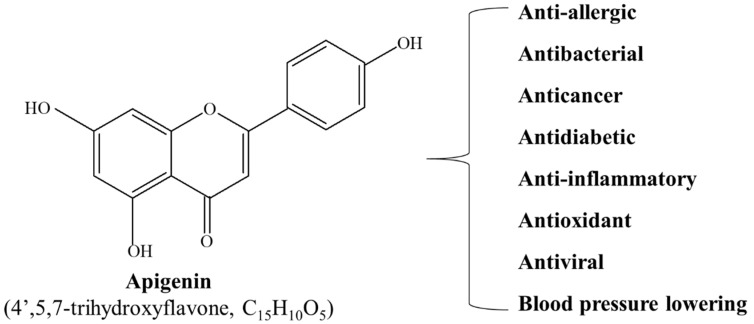
Molecular structure and physiological function of apigenin.

**Table 1 ijms-23-03757-t001:** Molecular targets of apigenin-induced apoptosis.

Cancer/Cell Lines	Up-Regulation	Down-Regulation	Refs.
** *Bladder* **			
T24	PARP cleavage, caspase-3, -7 and -9 cleavage, Bax, Bak, Bad, p–p53, p53, p21, p27, Cyt *c* (cytosol)	p-Akt, PDK, PI3K, Bcl-2, Bcl-xL, cyclin A, B1, and E, CDK2, Cdc2, Cdc25c, Bcl-xL, Mcl-1, Cyt *c* (mitochondrial)	[68,69]
RT112	PARP cleavage		[70]
** *Breast* **			
SK-BR-3	p53, p21, Bax, Cyt *c*, caspase-8 and -3, PARP, DFF45 cleavage, p27	cyclin A, B, D, and E, CDK1, p-JAK, p-STAT3, VEGF, cyclin D1 and D3, CDK4	[71,72,73]
MDA-MB-231 and MDA-MB-231 xenograft	p-p53 (Ser-15), p21, Bax, PARP cleavage, IκBα, caspase-3 and -7, FOXO3a, p27, Cyt *c*	Bcl-xL, cyclin B1, Bcl-2, PI3K, PKB/AKT	[74,75,76,77]
MCF-7	p53, p-p53 (Ser-15), p21, caspase-8 and PARP cleavage, ROS, Cyt *c*, caspase-3, DFF45 cleavage, p27, FOXO3a	p-MDM2, p-JAK1, p-STAT3, NF-κB/p65, p-IκBα, cyclin D1 and D3, CDK4, PI3K, PKB/AKT	[72,77,78,79,80]
BT-474	caspase-8 and -3, PARP, Cyt *c*, DFF45 cleavage, p27	p-JAK1, p-JAK2, p-STAT3, VEGF, HIF-1α, cyclin D1 and D3, CDK4	[72,81]
Hs578T	FOXO3a, p21, p27, PARP, Cyt *c* release	PI3K, PKB/AKT	[77]
MDA-MB-453	caspase-3, -6, -7, -8, and PARP cleavage, Cyt *c* release, DFF45 cleavage, p27	procaspase-9, p-JAK2, p-STAT3	[72,82,83]
T47D	caspase-3 and PARP cleavage, Bax	Bcl-2, Bcl-xL	[75]
HBL-100	Cyt *c*, caspase-3, DFF45 cleavage, p27	cyclin D1 and D3, CDK4	[72]
** *Cervical* **			
HeLa	p53, p21, caspase-2 and -3, Fas, mitochondrial redox impairment, PARP, ROS, AIF, Endo G, Cyt *c*	Bcl-2, MMP, superoxide dismutase	[84,85,86]
SiHa, CaSki, and C-33A	mitochondrial redox impairment, ROS	MMP	[85]
*Colon*			
HCT116	p21, p53, NAG-1, Bim-EL, Bim-L, PARP cleavage	cyclin B1, Cdc2, Cdc25c, procaspase-3, -8, and -9, Mcl-1, Bcl-xL, STAT3, p-AKT, p-ERK	[87,88,89,90]
LoVo	p21, NAG-1		[88]
DLD-1	PARP cleavage	Mcl-1, p-AKT, p-ERK, Bcl-xL, Mcl-1, STAT3	[87,90]
SW480		Cdc2, cyclin B1	[91]
HT-29	Bax, PARP cleavage, caspase-3 and -8	Cdc2, Bcl-2, m-TOR/PI3K/AKT, Bcl-xL, Mcl-1, STAT3, caspase-3 and -8, cyclin D1	[90,91,92,93]
Caco-2		Cdc2	[91]
COLO320	PARP cleavage	Bcl-xL, Mcl-1, STAT3	[90]
** *Esophageal* **			
KYSE-510	p21, PIG3, p63, p73, caspase-3 and -9, Bax	cyclin B1, Bcl-2	[94]
Eca-109 and KYSE-30	PARP cleavage, caspase-8	IL-6, VEGF	[95]
** *Gastric* **			
HGC-27 and SGC-7901	Bax, Bcl-2, caspase-3	MMP	[96]
** *Glioblastoma* **			
U-1242MG	PARP cleavage	MAPK, AKT, mTOR, Bcl-xL	[97]
T98G and U-87MG	p-p38 MAPK, c-Jun1, caspase-3, -8, and -9, Bax, tBid, Smac (cytosol), SBDP, CAD (nuclear)	ROS, MMP, Bcl-2, Cyt *c* (mitochondrial), Smac (mitochondrial), calpastatin, ICAD	[98]
***Head* and *Neck***			
SCC-25	TRAIL, TRAIL-R1, and -R2, Fas, TNF-α, TNF-R1 and -R2, Bax, caspase-3	Bcl-2	[99]
** *Melanoma* **			
A375 and C8161	Cyt *c* release, Bax, Apaf-1, caspase-3, -9, and PARP cleavage	Bcl-2, Cyt *c* (mitochondrial), p-ERK1/2, p-AKT, p-mTOR	[50,100,101]
** *Leukemia* **			
THP-1	caspase-3 activity, p-p38, p-ERK, PKCδ activity, p-ATM	caspase-9 activity, p-H2AX	[102,103]
U937	caspase-3, -7, -9, and PARP cleavage, p-JNK, Bcl-2 cleavage	hTERT, c-Myc, Mcl-1, p-AKT, AKT, p-Bad, p-mTOR, p-GSK3β, JNK, Mcl-1, Bcl-2	[104,105]
HL60	p-Cdc2, p-p38, caspase-3, -8, and PARP cleavage	PI3Kp85, p-AKT, p-GSK3β, p-JAK2, p-Src, p-STAT3	[106,107]
TF-1	LMWPTP	CDK6, p-Src, p-JAK2, p-SHP2, p-STAT3 and 5, p-p70S6K	[106]
** *Liver* **			
Huh-7	caspase-3, -8, and -9 cleavage, PARP, Bax/Bcl-2 ratio		[108,109]
HepG2	caspase-3, -7, -8, -9, and -10, Bid, p21, p16, PARP cleavage, Bax, DR5, ROS, TNF-α, IFN-γ	Bcl-2, PI3K/AKT/mTOR, p-LRP6, Skp2	[48,110,111,112,113,114]
Hep3B	DR5, ROS, caspase activation		[115]
SK-HEP-1	ROS, caspase 3, PARP	MMP, Bcl-2	[116]
BEL-7402 and BEL-7402 xenograft	ROS, caspase 3, PARP	MMP, Bcl-2, Nrf2	[116,117]
** *Lung* **			
A549	p21, Cyt *c* release, Bax, p53, p-p53, Wee1, Chk2, Bid, GRP78, caspase-3, -9, and PARP cleavage, GADD153, AIF, MAPK, DR4, DR5	XIAP, Bcl-2, MMP, cyclin B, Cdc25c, procaspase-8, Bcl-xL, NF-κB, ERK, AKT, Cyt *c* (mitochondrial)	[81,100,118,119,120]
H460	p21, Bax, FasL, p53, AIF, Cyt *c*, caspase-3, GRP78, GADD153	XIAP, Bcl-2, Bid, procaspase-8	[118,121,122]
H1299	MAPK, DR4, DR5, Bax, Bad	Bcl-xL, Bcl-2, NF-κB, ERK, AKT	[120]
** *Diffuse large B-cell lymphoma* **			
U2932 and OCI-LY10	caspase family, PARP cleavage	Bcl-xL, PI3K/mTOR, p-GS3K-β, MCL-X, p38, p-p65, p-AKT	[123]
** *Mesothelioma* **			
MM-B1, H-Meso-1 and MM-F1	Bax/Bcl-2 ratio, p53, caspase-8, -9, and PARP-1 cleavage	p-ERK1/2, p-JNK, p-p38 MAPK, p-AKT, c-Jun, p-c-Jun, NF-κB nuclear translocation	[124]
** *Multiple myeloma* **			
U266 and RPMI 8226	PARP cleavage	p-STAT3, p-ERK, p-AKT, NF-κB, Mcl-1, Bcl-2, Bcl-xL, XIAP, survivin	[125]
** *Neuroblastoma* **			
SK-N-DZ, SK-N-BE2, SK-N-DZ and SK-N-BE2 xenograft	caspase-3, -8, and PARP cleavage, Bax, Bid, tBid, calpain, ICAD fragment, p21, Noxa, PUMA, p53, ICAD, SBDP	N-Myc, E-cadherin, Notch-1, hTERT, PCNA, Smac, survivin, SBDP, Bcl-2, Mcl-1	[126,127,128,129]
NUB-7	PARP cleavage, p53 (NE), p21, Bax, p-ERK		[130]
IMR-32	Bax, Noxa, PUMA, p53, caspase-3, ICAD	Bcl-2, Mcl-1	[128]
** *Oral* **			
SCC-25	TRAIL, TRAIL-R1 and -R2, Fas, TNF-α, TNF-R1 and -R2, Bax, caspase-3	cyclin D1 and E, CDK1	[99]
** *Osteosarcoma* **			
U-2 OS	Bax, PARP cleavage, p53, AIF	procaspase-3, -8, and -9, GADD153 (NE)	[131]
** *Ovarian* **			
SKOV-3	caspase-3 and -9, Bax, Bcl-2, COX-2, ROS		[132,133,134]
A2780 and OVCAR-3	ROS, MDA, caspase-3 and -9		[133]
** *Pancreatic* **			
BxPC-3	Ac-p53, p21, PUMA, Cyt *c* release, caspase-3 cleavage	Bcl-xL/p53 interaction, Bcl-xL/PUMA interaction, cyclin B1, Bcl-2, XIAP, p-GSK3β, NF-κB/p65 (NE)	[135,136,137]
MIA PaCa-2	Ac-p53, p21, PUMA, Cyt *c* release, PARP cleavage	Bcl-xL/p53 interaction, Bcl-xL/PUMA interaction	[135,138]
PANC-1	Cyt c release, caspase-3 cleavage	cyclin B1, XIAP, p-GSK3β, NF-κB/p65 (NE)	[136]
** *PEL* **			
BC3, BCBL1, and B	p53	STAT3, ROS	[139]
** *Prostate* **			
22Rv1 and 22Rv1 xenograft	p53, p-p53, p21, p14, Cyt *c* release, Bax, Apaf-1, caspase-3, -8, -9, and PARP cleavage	MDM2, MMP, Bcl-2, Bcl-xL, p-IKKα, NF-ĸB/p65, PCNA, HDAC1 and 3, Bcl-2	[140,141,142]
PC-3 and PC-3 xenograft	caspase-3, -9, and PARP cleavage, Bax, Bad, Ku70, Cyt *c* release, p27, p21	XIAP, cIAP-1, -2, Bcl-2, Bcl-xL, survivin, HDAC1, procaspase-3, -7, and -9, cyclin D1, p-IKKα, NF-ĸB/p65, PCNA, ER-β, PSMA5, PLK-1, HDAC1, and 3, Bcl-2	[141,142,143,144,145,146,147]
LNCaP	p21, p27, Bax, PARP cleavage, Cyt *c* release	cyclin D1, D2, and E, CDK2, 4, and 6, Bcl-2, procaspase-3, -8, and -9, NF-κB/p65, PLK-1	[51,145,147]
DU145	caspase-3, -9, and PARP cleavage, DR5, Cyt *c* release	XIAP, cIAP-1 and -2, survivin, procaspase-3, -7, and -9	[143,145,148]
** *Renal* **			
ACHN, 786-O, and Caki-1	p53, Bax, caspase-3 and -9		[149]
** *Thyroid* **			
FRO	c-Myc, Bid, Fas, p-p53, caspase-3 and PARP cleavage	Bcl-2, p27, p21	[150]

AIF, apoptosis-inducing factor; Apaf-1, apoptotic protease activating factor-1; ATM, ataxia telangiectasia mutated; Bad, Bcl-2-associated death promoter; Bax, Bcl-2 associated X protein; Bcl-2, B-cell lymphoma-2; Bcl-xL, B-cell lymphoma extra-large; Bid, BH3-interacting-domain death agonist; Bim-EL, Bcl-2-interacting mediator of cell death (Bim)-extralong; Bim-L, Bim-long; CAD, caspase-activated DNase; Cdc2, cell division control protein 2; Cdc25c, cell division cycle 25c; CDK, cyclin-dependent kinase; Chk2, checkpoint kinase 2; cIAP, cellular inhibitor of apoptosis protein; COX-2, cyclooxygenase-2; Cyt *c*, cytochrome *c*; DFF45, DNA fragmentation factor 45; DR4, death receptors 4; DR5, death receptors 5; Endo G, endonuclease G; ER-β, estrogen receptor-beta; ERK, extracellular signal-regulated protein kinases; FasL, apoptosis stimulating fragment (Fas) ligand; FOXO3a, forkhead box O3a; GADD153, growth-arrest- and DNA-damage-inducible gene 153; GRP78, glucose-regulated protein 78; GSK-3β, glycogen synthase kinase-3 beta; H2AX, histone H2A, X; HDAC, histone deacetylase; hTERT, human telomerase reverse transcriptase; HIF-1α, hypoxia-inducible factor 1 alpha subunit; IκBα, nuclear factor of kappa light polypeptide gene enhancer in B-cells inhibitor alpha; JAK, Janus family of tyrosine kinase; JNK, c-Jun N-terminal kinases; LMWPTP, low-molecular-weight protein tyrosine phosphatase; MAPK, mitogen-activated protein kinase; Mcl-1, myeloid cell leukemia-1; MDA, malondialdehyde; MDM2, mouse double minute 2; mTOR, mammalian target of rapamycin; MMP, mitochondrial membrane potential; NAG-1, nonsteroidal anti-inflammatory drug (NSAID)-activated gene-1; NE, nuclear extract; Nrf2, nuclear factor erythroid 2-related factor 2; NF-κB, nuclear factor kappa-light-chain-enhancer of activated B cells; p70S6K, 70-kDa ribosomal protein S6 kinase; PARP, poly(ADP-ribose) polymerase; PCNA, proliferating cell nuclear antigen; PDK, phosphoinositide-dependent protein kinase; PEL, primary effusion lymphoma; PI3K, phosphoinositide 3-kinase; PIG3, p53 induced gene 3; PKB, protein kinase B; PKC, protein kinase C; PUMA, p53-upregulated modulator of apoptosis; ROS, reactive oxygen species; SBDP, spectrin breakdown product; SMAC, second mitochondria-derived activator of caspases; STAT, signal transducer and activator of transcription; TNFR, TNF receptor; TNF-α, tumor necrosis factor alpha; TRAIL, TNF-related apoptosis-inducing ligand; TRAIL-R, TRAIL receptor; VEGF, vascular endothelial growth factor; XIAP, X-linked inhibitor of apoptosis protein. Adapted in part from Sung, B.; Kim, N.D. Apigenin and Naringenin; Nova Science Publishers, Inc.: New York, NY, USA, 2015; pp. 75–106.

**Table 3 ijms-23-03757-t003:** Molecular targets of apigenin-induced necroptosis.

Cancer/Cell Lines	Up-Regulation	Down-Regulation	Refs.
** *Mesothelioma* **			
MSTO-211H and H2452	ROS, γ-H2AX, p-ATM, p-ATR, p-CHK1, p-CHK2, Bax, caspase-3 and PARP cleavage, p-MLKL, p-RIP3, Bax/Bcl-2 ratio	MMP, ATP, Bcl-2	[193]
** *Pancreatic* **			
AsPC-1	p-ATM, γ-H2AX, p-p53, Bim, Bid, Bax, PARP cleavage, caspasae-3, -8, and -9, Cyt *c*, AIF1, p62, LC3B, p-MLKL, p-RIP	Bcl-2	[194]

AIF1, apoptosis-inducing factor; ATM, ataxia telangiectasia mutated kinase; ATP, adenosine triphosphate; ATR, ataxia telangiectasia and Rad3-related kinase; Bax, Bcl-2-associated X protein; Bid, BH3 interacting-domain death agonist; Bim, Bcl-2 interacting mediator of cell death; Cyt *c*, cytochrome *c*; H2AX, H2A histone family member X; MLKL, mixed-lineage kinase domain-like pseudokinase; MMP, mitochondrial membrane potential; PARP, poly(ADP-ribose) polymerase; RIP3, receptor-interacting protein 3; ROS, reactive oxygen species.

**Table 4 ijms-23-03757-t004:** Molecular targets of apigenin-induced ferroptosis.

Cancer/Cell Lines	Up-Regulation	Down-Regulation	Refs.
** *Lung* **			
A549	ROS, COX-2, p53, MDA, Bax, caspase-3 and -8, Cyt *c*	GPX4, FTH1, SOD, Bcl-2	[207]
** *Multiple Myeloma* **			
HEK293	caspase-3 and -9, p38, JNK, LC3-II, Beclin-1, ROS	AKT, MMP, STAT1, COX-2, iNOS	[176]
NCI-H929	LC3-II, Beclin-1, ROS	MMP	[208]
** *Neuroblastoma* **			
SH-SY5Y	GPX4	MMP	[206]

Bax, Bcl-2-associated X protein; Cyt *c*, cytochrome *c*; COX-2, cyclooxygenase-2; FTH1, ferritin heavy chain 1; GPX4, glutathione peroxidase; iNOS, inducible nitric oxide synthase; JNK, c-Jun N-terminal kinases; MDA, malondialdehyde; MMP, mitochondrial membrane potential; ROS, reactive oxygen species; SOD, superoxide dismutase; STAT, signal transducer and activator of transcription.

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
