# Peer review of "Role of Induced Programmed Cell Death in the Chemopreventive Potential of Apigenin"

_ijms, 2022, doi:10.3390/ijms23073757_

Round 1
Reviewer 1 Report
Overall, a nice manuscript reviewing the effect of Apigenin on the cellular apoptosis pathway in the context of anti-cancer therapeutic. One minor concern is the autophagy-related section. Since the title specifically mentions the 'programmed cell death', the authors can consider removing the autophagy section from the manuscript.
Author Response
Response to Reviewer 1 Comments for ijms-1628926
Dear reviewers:
Thank you for your letter and for the reviewers’ comments concerning our manuscript entitled “Role of Induced Programmed Cell Death in the Chemopreventive Potential of Apigenin ” (ID: ijms-1628926). These comments are all valuable and very helpful for revising and improving our paper, as well as the important guiding significance to our researches. We have studied comments carefully and have made corrections which we hope meet with approval. Revised portions are marked in red on the paper. The main corrections in the paper and the response to the reviewer’s comments are as flowing:
Point 1: Overall, a nice manuscript reviewing the effect of Apigenin on the cellular apoptosis pathway in the context of anti-cancer therapeutic. One minor concern is the autophagy-related section. Since the title specifically mentions the 'programmed cell death', the authors can consider removing the autophagy section from the manuscript.
Response 1: Thanks for the comment. Autophagy has been reported as programmed cell death in many previous papers [1-7]. According to a paper by Tsujimoto et al. [4], the so-called 'autophagic programmed cell death', a process related to autophagosome and autolysosome, is reported.
Schweichel and Merker published a morphological characterization system for classifying cell death into types I, II and III in prenatal tissues treated with various embryotoxic substances in 1973. Type II cell death, often referred to as autophagy-dependent cell death, is accompanied by the formation of large-scale autophagic vacuolizaion-containing cytosolic material and organelles [5].
The current classification system for cell death has been updated since 2005 by the Nomenclature Committee on Cell Death (NCCD), which has formulated guidelines for the definition and interpretation of all aspects of cell death [6]
In 2018, the NCCD stated that the fully physiological form of regulated cell death (RCD) is commonly referred to as programmed cell death. Various RCD have been described including apoptosis (extrinsic and intrinsic), autophagy-dependent cell death, necroptosis, mitochondrial permeability transition-driven necrosis, ferroptosis, pyroptosis, parthanatos, entotic cell death, NETotic cell death, lysosome-dependent cell death, and immunogenic cell death. Autophagy is also referred to as type II cell death [2].
Many papers classify regulated cell death according to the NCCD [3,6,7]
Also, according to a recent paper by Yan et al. [7], cell death was classified as programmed cell death or non-programmed cell death, and autophagy was classified as programmed cell death. In addition, programmed cell death was classified into two categories, programmed apoptotic cell death and programmed non-apoptotic cell death. Autophagy, along with entosis, methuosis, and paraptosis, belongs to programmed non-apoptotic cell death that produces vacuole.
- Tsujimoto, Y.; Shimizu, S. Another way to die: autophagic programmed cell death. Cell Death Differ 2005, 12 Suppl 2, 1528-1534, doi:10.1038/sj.cdd.4401777.
- Galluzzi, L.; Vitale, I.; Aaronson, S.A.; Abrams, J.M.; Adam, D.; Agostinis, P.; Alnemri, E.S.; Altucci, L.; Amelio, I.; Andrews, D.W.; et al. Molecular mechanisms of cell death: recommendations of the Nomenclature Committee on Cell Death 2018. Cell Death Differ 2018, 25, 486-541, doi:10.1038/s41418-017-0012-4.
- Hu, X.M.; Li, Z.X.; Lin, R.H.; Shan, J.Q.; Yu, Q.W.; Wang, R.X.; Liao, L.S.; Yan, W.T.; Wang, Z.; Shang, L.; et al. Guidelines for regulated cell death assays: A Systematic Summary, A categorical comparison, a prospective. Front Cell Dev Biol 2021, 9, 634690, doi:10.3389/fcell.2021.634690.
- Ouyang, L.; Shi, Z.; Zhao, S.; Wang, F.T.; Zhou, T.T.; Liu, B.; Bao, J.K. Programmed cell death pathways in cancer: a review of apoptosis, autophagy and programmed necrosis. Cell Prolif 2012, 45, 487-498, doi:10.1111/j.1365-2184.2012.00845.x.
- Schweichel, J.U.; Merker, H.J. The morphology of various types of cell death in prenatal tissues. Teratology 1973, 7, 253-266, doi:10.1002/tera.1420070306.
- Tang, D.; Kang, R.; Berghe, T.V.; Vandenabeele, P.; Kroemer, G. The molecular machinery of regulated cell death. Cell Res 2019, 29, 347-364, doi:10.1038/s41422-019-0164-5.
- Yan, G.; Elbadawi, M.; Efferth, T. Multiple cell death modalities and their key features. World Acad Sci 2020, 2, 39-48.
Therefore, it would be better to keep the current manuscript describing autophagy as a type of programmed cell death.
We tried our best to improve the manuscript and made some changes in the manuscript. These changes will not influence the content and framework of the paper. And here we did not list the changes but marked them in red in the revised paper.
We appreciate for Editors/Reviewers’ warm work earnestly and hope that the correction will meet with approval. Once again, thank you very much for your comments and suggestions.
Reviewer 2 Report
Cancer diseases are still a common, more and more knowing entity, but with complications in standard treatment at the same time, with relatively low survival rate. Especially the malignant neoplasms are a hard nut to crack in case of surgical resection and radio- or chemotherapy. This is why scientists all over the world search for new drug treatments leading to complete regression of the disease. Special attention is paid to the therapy based on natural compounds, especially flavonoids which give the hope for better maintaining the cancers and improve the recovery of patients. The standard treatment or modern molecular targeted therapies may carry some serious side effects, and it is believed the plants material could reduce those with the improvement of cancer cell elimination upon e.g. programmed cell death induction.
The Authors in their work present valuable knowledge about one of the best-known compounds of plant origin, which is apigenin.
They describing its role in the cancer cells elimination and the molecular basics of those mechanisms.
The work is properly divided for the chapters starting with the Introduction, than the apigening flavonoid is well characterized, what helps with the understanding of its mechanisms of action upon cacner cells. Special attention the authors pay to programmed cell death (PCD) mechanisms, especially the apoptosis and authophagy, which are the most important pathways studying in cancer cell elimination. However, there are interactions between them and those mechanisms are more complicity. The Authors could describe some of specific circumstances such as a Bcl-2 and beclin-1 complexes occurance and the correlation between the intracellular survival pathways overexpression and different types of death induction. Moreover, authophagy is ambiguous process and may lead to surivive of cancer cells or its partial elimination with the inflammatory induction, that is why athe apoptotic effect is more desirable in cancer treatment. The different mechanisms may lead to cancer cell survival (especially the autophagy one) instead of their elimination, what decreases the disease regression. Some of those informations are presented in the big and very proper table (Table 1) with a huge scientific value, but authors could more explain it in the text. Maybe not every of them of course, but some the most important once e.g. the correlation between the caspases and beclin-1 protein. The Reviewer is satisfied with the explanation of abbreviations.
The discriptions and exlaining of the DNA damage mechanisms are very important in the tumor growth genesis, and the Authors described them properly.
However, some minor mistakes within the manuscript were found by the Reviewer:
- Introduction - line 28 - double space between "and" and "molecular"
- Involving the usege of "natural, synthetic, or biological agents" is not only for inhibition and prevention of early stages of cancer diseases, and the natural compounds could be use as a supportive or adjuvants during the standard chemotherapy reducing the side effects.
- Line 56 - "The compound has very low toxicity" the Reviewer suggest change this sentence, the toxicity of apigenin is lower upon normal than cancer cells but it could still occur, and there are some reports which confirm that in literature.
- The structural picture of apigening (Figure 1) could have a better resolution (dpi).
- Line 90 - "cell cycle suspension" in molecular targeted treatment the cel cycle arrest is commonly used form, the Reviewer suggest to chagne it.
- Line 101 - there are some double spaces.
- 5.1. Apoptosis - the sentence
"disruption of this balance (e.g., uncontrolled apoptosis) has been implicated in a variety of human diseases, including cancer" suggest the cancer progression could be increase upon uncontrolled apoptosis, and it is known the cancer cells involve many mechnisms for avoiding programmed cell death induction, especially apoptosis for survive. The Reviewer kindly request for the extention of Authors idea or changing this sentence to clarify. - 6.1. Autophagy - line 271, 276 - double space occour.
- 6.3. Induction of Autophagy by Apigenin - line 333 - it is double space.
However, the above errors do not affect the substantive and scientific value of this work presented to me for review, and the Revierwer recommend it to further publication process after minor corrections.
Author Response
Response to Reviewer 2 Comments for ijms-1628926
Dear reviewer:
Thank you for your letter and for the reviewers’ comments concerning our manuscript entitled “Role of Induced Programmed Cell Death in the Chemopreventive Potential of Apigenin” (ID: ijms-1628926). These comments are all valuable and very helpful for revising and improving our paper, as well as the important guiding significance to our researches. We have studied comments carefully and have made corrections which we hope meet with approval. Revised portions are marked in red on the paper. The main corrections in the paper and the response to the reviewer’s comments are as flowing:
Point 1: Introduction - line 28 - double space between "and" and "molecular"
Response 1: Thanks for the comment. We are very sorry for the mistakes. We have corrected the manuscript accordingly.
Point 2: Involving the usage of "natural, synthetic, or biological agents" is not only for inhibition and prevention of early stages of cancer diseases, and the natural compounds could be used as a supportive or adjuvants during the standard chemotherapy reducing the side effects.
Response 2: Thanks for your advice. We have revised the manuscript accordingly.
Point 3: Line 56 - "The compound has very low toxicity" the Reviewer suggest change this sentence, the toxicity of apigenin is lower upon normal than cancer cells but it could still occur, and there are some reports which confirm that in literature.
Response 3: Thank you for your comments. According to the reference, side effects of apigenin are reported to be low.
Point 4: The structural picture of apigenin (Figure 1) could have a better resolution (dpi).
Response 4: We are very sorry for the low quality of the figures, and we have revised manuscript accordingly.
Point 5: Line 90 - "cell cycle suspension" in molecular targeted treatment the cell cycle arrest is commonly used form, the Reviewer suggest to change it.
Response 5: Thanks for the comment. We have revised some parts of the manuscript accordingly.
Point 6: Line 101 - there are some double spaces.
Response 6: Thanks. However, there is no double space occurrence.
Point 7: 1. Apoptosis - the sentence "disruption of this balance (e.g., uncontrolled apoptosis) has been implicated in a variety of human diseases, including cancer" suggest the cancer progression could be increase upon uncontrolled apoptosis, and it is known the cancer cells involve many mechanisms for avoiding programmed cell death induction, especially apoptosis for survive. The Reviewer kindly request for the extension of Authors idea or changing this sentence to clarify.
Response 7: Thanks for your advice. We have revised the manuscript accordingly.
Point 8:1. Autophagy - line 271, 276 - double space occour.
Response 8: We are very sorry for the mistakes. Thank you. We revised the manuscript accordingly.
Point 9: 3. Induction of Autophagy by Apigenin - line 333 - it is double space.
Response 9: We are very sorry for the mistakes. Thank you. We revised the manuscript accordingly.
We tried our best to improve the manuscript and made some changes in the manuscript. These changes will not influence the content and framework of the paper. And here we did not list the changes but marked them in red in the revised paper.
We appreciate for Editors/Reviewers’ warm work earnestly and hope that the correction will meet with approval. Once again, thank you very much for your comments and suggestions.
